# Metastatic colorectal adenocarcinoma tumor purity assessment from whole exome sequencing data

Noura Tbeileh[1,2¤a], Luika Timmerman[2]*, Aras N. Mattis[2,3,4], Kan Toriguchi![ORCID][1,2¤b], Yosuke Kasai[1,2¤c], Carlos Corvera[1,2], Eric Nakakura[1,2], Kenzo Hirose[1,2], David B. Donner[1,2], Robert S. Warren![ORCID][1,2☯]*, Eveliina Karelehto![ORCID][1,2☯]

1 Department of Surgery, University of California, San Francisco, San Francisco, California, United States of America, 2 UCSF Helen Diller Family Comprehensive Cancer Center, San Francisco, California, United States of America, 3 Department of Pathology, University of California, San Francisco, San Francisco, California, United States of America, 4 Liver Center, University of California, San Francisco, San Francisco, California, United States of America

☯ These authors contributed equally to this work.
¤a Current address: University of California, San Diego, San Diego, California, United States of America
¤b Current address: Department of Gastroenterological Surgery, Hyogo College of Medicine, Hyogo, Japan
¤c Current address: Department of Surgery, Kyoto University Graduate School of Medicine, Kyoto, Japan
* robert.warren@ucsf.edu (RSW); luika.timmerman@ucsf.edu (LT)

**Data Availability Statement:** All relevant data are within the paper. Whole-exome sequences of patients cannot be made publicly available for patient confidentiality reasons. This data is

## Abstract

Tumors rich in stroma are associated with advanced stage and poor prognosis in colorectal adenocarcinoma (CRC). Abundance of stromal cells also has implications for genomic analysis of patient tumors as it may prevent detection of somatic mutations. As part of our efforts to interrogate stroma-cancer cell interactions and to identify actionable therapeutic targets in metastatic CRC, we aimed to determine the proportion of stroma embedded in hepatic CRC metastases by performing computational tumor purity analysis based on whole exome sequencing data (WES). Unlike previous studies focusing on histopathologically pre-screened samples, we used an unbiased in-house collection of tumor specimens. WES from CRC liver metastasis samples were utilized to evaluate stromal content and to assess the performance of three in silico tumor purity tools, ABSOLUTE, Sequenza and PureCN. Matching tumor derived organoids were analyzed as a high purity control as they are enriched in cancer cells. Computational purity estimates were compared to those from a histopathological assessment conducted by a board-certified pathologist. According to all computational methods, metastatic specimens had a median tumor purity of 30% whereas the organoids were enriched for cancer cells with a median purity estimate of 94%. In line with this, variant allele frequencies (VAFs) of oncogenes and tumor suppressor genes were undetectable or low in most patient tumors, but higher in matching organoid cultures. Positive correlation was observed between VAFs and in silico tumor purity estimates. Sequenza and PureCN produced concordant results whereas ABSOLUTE yielded lower purity estimates for all samples. Our data shows that unbiased sample selection combined with molecular, computational, and histopathological tumor purity assessment is critical to determine the level of stroma embedded in metastatic colorectal adenocarcinoma.

deposited in the dbGaP repository (accession number phs003059), for restricted access for researchers who meet the dbGaP criteria for access to confidential data.

**Funding:** This work was supported by a University of California Cancer Research Coordinating Committee award #C21CR2154 to RSW (https://ucop.edu/research-initiatives/programs/crcc/index.html) and in part by a gift from the Edmund Wattis Littlefield Foundation to RSW. The funders had no role in study design, data collection and analysis, decision to publish, or preparation of the manuscript.

**Competing interests:** The authors have declared that no competing interests exist.

## Introduction

Tumors are heterogeneous mixtures of cancer cells and non-cancerous stromal elements, such as fibroblasts, endothelial and immune cells [1]. The term used to describe the proportion of malignant cells versus stroma within the tumor mass is purity. Histologically, low tumor purity i.e., high level of stroma embedded within the tumor mass, has been linked to poor prognosis in colorectal cancer (CRC) [2, 3]. Transcriptomic analyses and molecular subtyping of CRC also support the conclusion that an abundance of stromal cells within the tumor, particularly cancer-associated fibroblasts, correlates with worse survival [4]. Additionally, low tumor purity hinders genomic and transcriptomic characterization of the malignant cells, for example, by preventing accurate detection of somatic variants in cancer driver genes thus leading to false negative findings [5].

To determine tumor purity, pathologists examine hematoxylin and eosin (H&E) -stained tumor sections visually and estimate the fraction of malignant and stromal cells within the sample. Recently, computational tools employing data generated by molecular assays such as single nucleotide polymorphism arrays, DNA methylation, RNA-sequencing, and whole exome sequencing (WES) have been developed to assess tumor purity. Such *in silico* tools can be differentiated by the methods they use to infer purity e.g., copy number alterations (CNAs), somatic mutations, loss of heterozygosity signals, allelic fraction values, or deep learning models [6]. The computational tools strive to provide unbiased estimates of tumor purity, but the accuracy of such estimates have varied between tools depending on the method of inference used by the tool. Previous assessments of *in silico* purity tools have used public databases such as TCGA for their sample sets [6, 7]. While the TCGA CRC database is extensive, the samples in it have been histologically prescreened to omit those with tumor purity below 60% thus leading to biased sampling [8].

Here we aimed to overcome this sampling bias by investigating tumor purity in a cohort of randomly selected in-house colorectal adenocarcinoma liver metastasis (CRCLM) specimens. WES and estimated tumor purity in tissue specimens collected from chemotherapy naïve and treated patients, as well as with matching tumor organoid cultures, was determined and then evaluated using three *in silico* tumor purity tools: ABSOLUTE [9], Sequenza [10], and PureCN [11]. Overall, we observed lower tumor purity than was previously reported [8] for primary CRC samples, with a median computed purity below 50% across all CRCLM patient tumors and all computational methods used in this study. Variant allele frequencies (VAFs) of the pathogenic mutations found in patient tumors were consistent with the computational tumor purity estimates. As expected, organoid cultures were enriched for cancer cells with median estimated purity above 90% and high VAFs. Additionally, we found varying concordance among the *in silico* tools and between the computational and pathologist tumor purity estimates.

## Materials and methods

### Samples and consent

Eighteen patients underwent resection for colorectal adenocarcinoma liver metastasis at the University of California, San Francisco (UCSF). The research protocol was approved by the institutional review board of University of California San Francisco (IRB#10–05031). This study was exempt from informed consent as only excess archival patient tissue was collected and as samples were subsequently anonymized.

No minors were involved in these studies. One half of the excess tissue sample from resected liver metastases and adjacent normal liver was snap frozen by transfer to a Dewar

**Table 1. Overview of patient colorectal adenocarcinoma liver metastasis samples.**

| Patient ID | Primary tumor location | Chemotherapy | Matched normal DNA | Tumor organoids | Sample type used for tumor DNA | Pathologist assessment |
|---|---|---|---|---|---|---|
| CR1106 | Colon | Yes | Yes | PDXO | Snap frozen | FFPE |
| CR1107 | Rectum | Yes | Yes | PDO | FFPE | FFPE |
| CR1116 | Colon | Yes | Yes | PDXO | Snap frozen | FFPE |
| CR1119 | Colon | Yes | Yes | PDXO | Snap frozen | FFPE |
| CR1121 | Colon | Yes | Yes | PDO | Snap frozen | FFPE |
| CR1123 | Rectum | Yes | Yes | PDXO | Snap frozen | FFPE |
| CR550 | Colon | Yes | n/a | n/a | Snap frozen | FFPE |
| CR611 | Colon | Yes | n/a | n/a | Snap frozen | FFPE |
| CR623 | Colon | Yes | n/a | n/a | Snap frozen | n/a |
| CR644 | Colon | No | n/a | n/a | Snap frozen | FFPE |
| CR655 | Colon | Yes | n/a | n/a | Snap frozen | n/a |
| CR661 | Colon | No | n/a | n/a | Snap frozen | n/a |
| CR674 | Colon | Yes | n/a | n/a | Snap frozen | n/a |
| CR692 | Colon | No | n/a | n/a | Snap frozen | n/a |
| CR703 | Colon | Yes | n/a | n/a | Snap frozen | FFPE |
| CR704 | Colon | No | n/a | n/a | Snap frozen | n/a |
| CR719 | Colon | No | n/a | n/a | Snap frozen | FFPE |
| CR726 | Colon | No | n/a | n/a | Snap frozen | FFPE |

PDO; patient-derived organoids, PDXO; patient-derived xenograft organoids, FFPE; formalin-fixed paraffin-embedded, n/a; not available.

flask containing liquid nitrogen, transported to the laboratory, and stored at −80˚C. The remaining portion of the tumor specimen was used to generate a mouse xenograft and to isolate tumor organoids. All animal experiments were carried out by members of the UCSF Preclinical Therapeutics Core Facility in accordance with the University of California San Francisco animal care and use committee (IACUC# AN179937-03A). Animals were anesthetized using isoflurane (2–4%) during SC PDX implantation and, at endpoint, euthanized by asphyxiation with CO2 followed by cervical dislocation, as recommended by the Panel on Euthanasia of the American Veterinary Medical Association." An overview of the samples used for this study is shown in Table 1.

## Organoid culture

Tumor organoids were generated either directly from patient tumor (patient-derived organoids, PDOs) or from the xenograft tumor (patient-derived xenograft organoids, PDXOs). For two patients xenografts did not take but organoid lines could be established directly from the patient tumors. Conversely, PDOs could not be generated directly from four patient tumor specimens and instead PDXOs were used. All organoids were generated by the method previously described by Kondo et al. [12] with some modifications. Briefly, patient or PDX tumor specimens were minced using a scalpel and digested using Liberase (Sigma) and DNase (Qiagen) in a Gentle Macs Octo Dissociator (Miltenyi) for 1 hour at 37 degrees C. Dissociated tumor suspension was then sequentially filtered, first through 500μm, 250μm and 100μm filters to remove undigested material, and then a 40μm filter was employed to retain small clusters of cells while allowing individual cells to pass through. The cell clusters were transferred to ultra-low attachment plates and cultured in organoid media (DMEM/F12 Glutamax (Gibco), 1x PenStrep/Glutamine (Invitrogen), 1x STEMPRO hESC SFM (Invitrogen), 0.1mM beta-mercaptoethanol, 8ng/mL bFGF (Invitrogen), 1.8%BSA (Invitrogen) and 2% growth factor

reduced Matrigel (Corning). DNA for whole exome sequencing was extracted from low passage organoids (p < 5) except for CR1107 PDOs for which additional passaging (p = 16) was necessary to obtain sufficient yield.

## Whole exome sequencing

RNA-free genomic DNA was extracted from fresh frozen tumor specimens and from organoids using the Macherey-Nagel NucleoSpin Tissue mini kit according to manufacturer instructions. DNA from patient CR1107 tissues was extracted from formalin fixed paraffin embedded (FFPE) sections of tissue at Novogene Co., Ltd in Beijing, China. Whole exome capture and sequencing (WES) was performed at Novogene Co.at sequencing depth of 100X, Ltd. Briefly, genomic DNA was randomly sheared into short fragments of 180–280 bp. The fragments were end repaired, A-tailed, and further ligated with Illumina adapters. The fragments with adapters were PCR amplified, size selected, and purified. The prepared libraries were hybridized in buffer with biotin-labeled probes, and magnetic beads with streptavidin captured the exons of genes. Subsequently, non-hybridized fragments were washed out and probes were digested. The captured libraries were enriched by PCR amplification. Library quality was assessed using Qubit and real-time PCR for quantification, and bioanalyzer for size distribution detection. Quantified libraries were pooled and sequenced on Illumina platform with PE150. Burrows-Wheeler Aligner (BWA) mapped the paired-end clean reads to the reference genome (GRCh38) [13]. SAMtools sorted and indexed the original BAM file followed by Picard to mark duplicate reads [14]. The WES data from this study has been deposited in NIH dbGaP repository with accession number: phs003059.

## Variant calling

For detecting pathogenic variants in patient tumors, single nucleotide polymorphisms were called by using GATK's HaplotypeCaller from BAM files and annotated by ANNOVAR [15, 16]. These variants were then filtered down to variants determined to be pathogenic in the ClinVar database [17]. For the *in silico* tumor purity tools described below, somatic and germline variants were called using GATK4's Mutect version 2.2 according to GATK Best Practices. Tumor-only samples were called with af-only-gnomad.hg38.vcf.gz as the germline resource (from Broad Institute's Google Cloud Bucket) with the flags—genotype-germline-sites true—genotype-pon-sites true to keep germline mutations in the output VCF. Matched tumor samples and organoids were run using the same flags with their additional matched normal BAM file.

## ABSOLUTE

ABSOLUTE infers purity from relative copy number profiles from the provided segmentation file input. Ambiguous cases are resolved through pre-computed statistical models of cancer karyotypes based on a diverse sample reference collection. This algorithm also attempts to account for copy number alterations and point mutations in tumor subclones [9]. A segmentation file for each clinical and organoid sample with a matched normal liver sample was produced using GATK4's ModelSegments CNA workflow. This file contains total segmented copy ratios for the tumor sample. ABSOLUTE used this file as input and was run with the default parameters. For each sample, purity was estimated with the max.non.clonal parameter set to the default 5%, 30% and 50% to account for tumor heterogeneity. Purity for each run was accepted as the maximum log-likelihood solution.

### Sequenza

Sequenza performs allele-specific segmentation before applying a probabilistic model to segmented data, taking into account the average sequencing depth ratio of tumor versus normal and B allele frequency and estimating model parameters through a maximum *a posteriori approach* to infer purity and ploidy [10]. Tumor, organoid, and matched normal BAM files were analyzed using the workflow described in the Sequenza User Guide. Input files were preprocessed using sequenza-utils, the Python library accompanying the Sequenza R-package. The preprocessed files were analyzed using the Sequenza R-package. Purity for each sample was accepted as the first solution (of "cellularity") in the confints_CP.txt file output which was determined through a maximum likelihood estimation and the 95% confidence interval.

### PureCN

PureCN employs a likelihood model on segmented data that identifies artifacts caused by incorrect read alignment or contamination of DNA from other individuals, incorporates the important information provided by somatic point mutations from VCF input, uses copy number and SNV information jointly, and supports uneven tiling of targets across the genome to give the best purity estimate [11]. PureCN was used to call the purities of tumor-only samples using its tumor-only mode according to PureCN's best practices. Data was segmented through PureCN's internal segmentation method. GC-normalized coverages were calculated for all samples using PureCN's script Coverage.R. A normal database was then built using all normal sample coverages using NormalDB.R. Then PureCN's main script was run to infer purity taking as input the tumor sample's normalized GC coverage file, VCF from Mutect, normal database, and baits interval file obtained from Agilent. Tumor and organoid samples with a matched normal were run using a similar workflow. The normal database for each run contained all normal samples except the one used in the matched run. PureCN's main script took as input the tumor or organoid normalized GC coverage file, the matched normal normalized GC coverage file, the normal database, VCF from Mutect, and baits interval file. Accepted purity for every sample was determined to be the maximum likelihood solution determined by PureCN for each sample.

### Pathologist estimate

For tumor samples with an available FFPE tumor specimen, a board-certified gastrointestinal pathologist from UCSF estimated the percentage of tumor cells within the tumor area of the tissue section. Additionally, proportions of necrosis and fibrosis were estimated within the tumor area. All estimates were based on 4μm thick H&E-stained sections of the specimens.

### Statistical analysis

The Spearman rank test was used to measure correlation between the variant allele frequencies and *in silico* tumor purity assay results. The mean tumor purity between chemotherapy naïve and treated patient specimens was compared by unpaired two-tailed t test with alpha level of 0.05. Data were analyzed using GraphPad Prism 9 (GraphPad Software Inc., La Jolla, CA).

## Results

We employed three *in silico* tools, ABSOLUTE, Sequenza and PureCN, to estimate tumor purity in CRCLM from 6 patients for which matching normal, tumor and tumor-derived organoid WES data was available. Results for the patient tumor samples and the matching organoids are displayed for each tool in Fig 1A and 1B. Based on the TCGA guidelines, we

| ABSOLUTE 30% | ABSOLUTE 50% | Sequenza | PureCN | |
|:---:|:---:|:---:|:---:|:---|
| 16 | 16 | 48 | 59 | **CR1106** |
| 25 | 25 | 37 | 38 | **CR1107** |
| 30 | 30 | 90 | 91 | **CR1116** |
| 57 | 57 | 72 | 72 | **CR1119** |
| 18 | 18 | 27 | 32 | **CR1121** |
| 19 | 19 | 20 | 23 | **CR1123** |

**A.**

| ABSOLUTE 30% | ABSOLUTE 50% | Sequenza | PureCN | |
|:---:|:---:|:---:|:---:|:---|
| 33 | 33 | 100 | 95 | **PDXO1106** |
| 36 | 36 | 100 | 56 | **PDO1107** |
| 35 | 35 | 100 | 95 | **PDXO1116** |
| 35 | 35 | 100 | 93 | **PDXO1119** |
| 100 | 100 | 100 | 95 | **PDO1121** |
| 30 | 98 | 100 | 51 | **PDXO1123** |

**B.**

| ABSOLUTE 30% | ABSOLUTE 50% | Sequenza | PureCN | |
|:---:|:---:|:---:|:---:|:---|
| 100 | 100 | 100 | 95 | **100% / 0%** |
| 26 | 26 | 58 | 58 | **75% / 25%** |
| 22 | 22 | 15 | 16 | **30% / 70%** |
| 21 | 21 | 10 | 15 | **15% / 85%** |

**C.**

### *In silico* tumor purity (%)

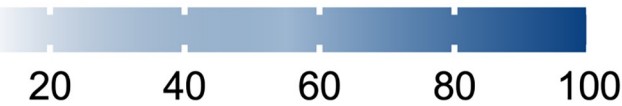

**Fig 1. In silico tumor purity assessment using whole-exome sequencing data and ABSOLUTE, Sequenza and PureCN tools.** ABSOLUTE was run with the max.non.clonal parameter set to either 30% or 50%. CRCLM tumors (**A**) and tumor-derived organoids (**B**) from 6 patients for whom matching normal DNA was available were analyzed. (**C**) Tumor purity estimates of samples with known ratios of normal and tumor DNA from patient CR1121. PDO; patient-derived tumor organoids, PDXO; patient-derived xenograft organoids.

consider purity less than 60% low and more than 60% high [20]. Median estimated purities for the patient tumors by all methods were low (ABSOLUTE 22%, Sequenza 42.5% and PureCN 48.5%) whereas estimates for the tumor-derived organoids were higher (ABSOLUTE 35%, Sequenza 100% and PureCN 93%). ABSOLUTE consistently produced lower purity estimates than the other two tools. Higher tumor purity was expected for the organoid samples as stromal components are typically lost during organoid generation [18, 19]. In fact, we used normal DNA from patient CR1121 and matching CR1121 tumor organoids to compare the three tools against a sample set with known ratios of normal and tumor DNA (Fig 1C). Organoid and normal DNA was mixed at following ratios (organoid% /normal%): 75/25, 30/70 and 15/85. All tools predicted the pure 100% organoid sample to have tumor purity of 95–100% whereas estimates for the mixed samples were variable. Sequenza and PureCN estimated the 75/25 sample to have purity of 58% while the ABSOLUTE result was lower at 26%. Purity estimates for the 30/70 and 15/85 samples from all tools were underestimates with results ranging between 20–10%. We subsequently employed the only tool of the three compatible with tumor only data, PureCN, to analyze additional CRCLM tumors from 12 patients for which no matching normal or organoid DNA was available. As shown in Fig 2, with few exceptions we again observed low median tumor purity (median 46.5%, range 17–89%).

A pathologist estimated the tumor purity of FFPE sections as the percentage of tumor cells within the tumor area. Representative H&E stained section of a CRCLM patient tumor (CR726) is shown in Fig 3. Pathologist tumor purity estimates for all available samples are shown in Table 2. Proportion (%) of necrosis and fibrosis within the tumor area was also

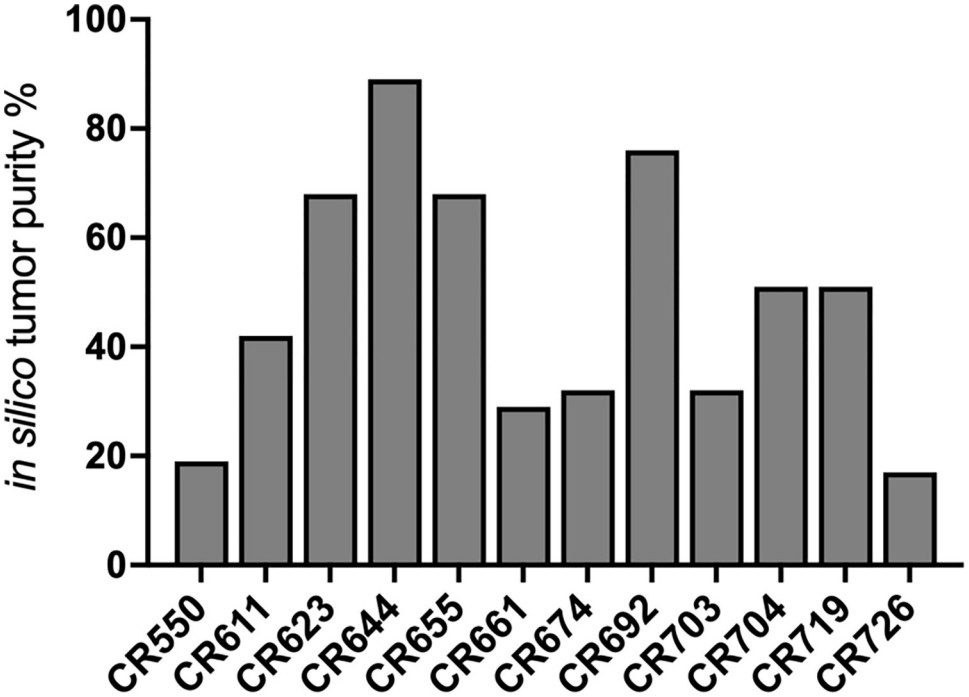

**Fig 2. PureCN tumor purity assessment of CRCLM patient tumors from 12 patients using tumor only data.**

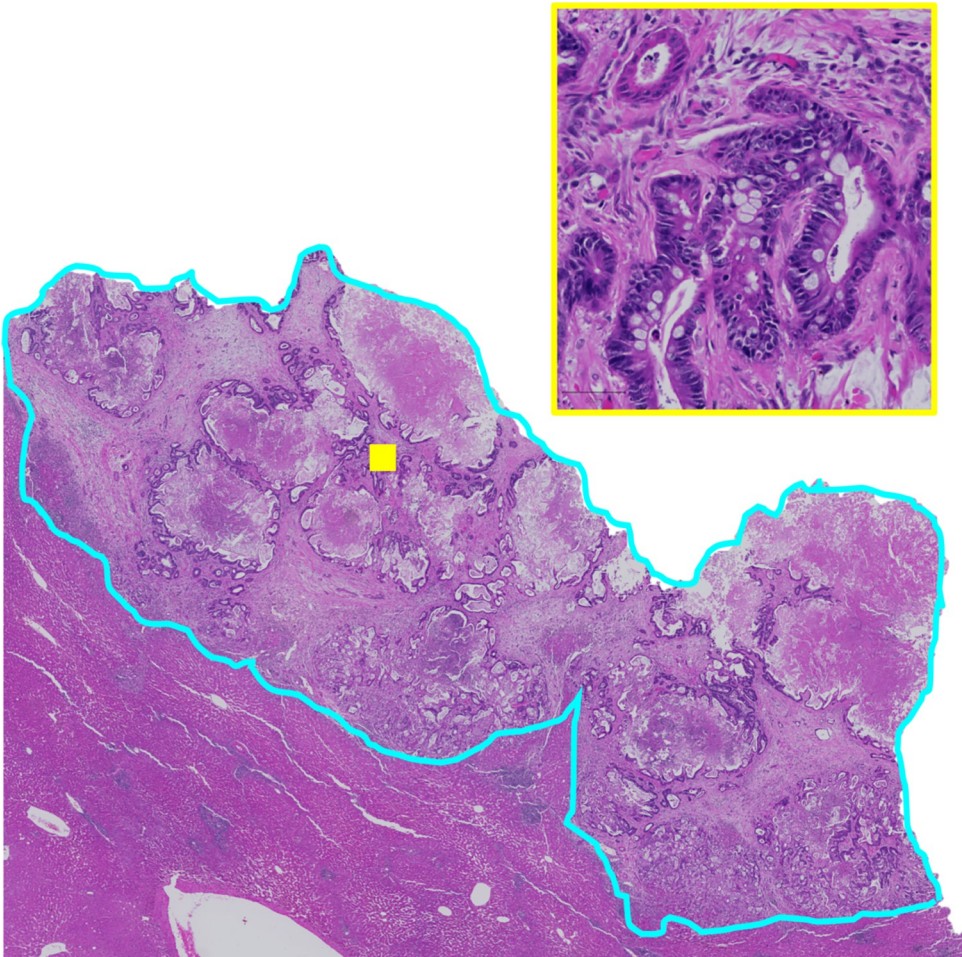

**Fig 3. Representative H&E stained section of the patient CR726 FFPE CRCLM tumor sample.** Tumor area is outlined with cyan and the zoomed in area within it in yellow.

assessed. *In silico* tumor purity results from the PureCN tool are also included in Table 2 for comparison. For approximately half of the samples, pathologist tumor purity estimates were similar to those obtained by *in silico* tools but for the remaining samples the estimates differed.

Tumor WES data from all 18 patients was screened for known pathogenic variants and the variant allele frequencies (VAF) were investigated. As shown in Fig 4A, relatively few pathogenic variants were detected and apart from patient tumors CR1116, CR692, and CR644, variants exhibited low VAFs. For example, APC and TP53 mutations across all tumors on average had VAFs of 19% and 26%, respectively. In contrast, the majority of the sequencing reads from tumor organoids harbored the same pathogenic variants as their parent tumors but with higher VAFs (Fig 4B). Furthermore, we found significant positive correlations between the VAF of the most prevalent pathogenic variant in each sample and the *in silico* tumor purity estimates from Sequenza and PureCN algorithms and from ABSOLUTE when run with max. non.clonal parameter set to 50% (Fig 5A–5D). However, there was no significant correlation between the pathologist tumor purity assessment and the VAFs (Fig 5E), nor between the pathologist purity assessment and the PureCN in silico purity results (Fig 5F).

Lastly, we stratified the tumor samples based on patient chemotherapy status prior to hepatectomy and compared the tumor purity estimates between chemotherapy naïve and treated

**Table 2. Pathologist estimate of tumor purity based on H&E stained sections of FFPE CRCLM tumor samples from 12 patients.**

| | Pathologist | | | In silico |
|---|---|---|---|---|
| Patient ID | Tumor purity [a] | Necrosis [b] | Fibrosis [b] | Tumor purity [c] |
| CR1106 | 20 | 25 | 10 | 59 |
| CR1107 | 60 | 50 | 10 | 38 |
| CR1116 | 65 | 70 | 5 | 91 |
| CR1119 | 70 | 10 | 10 | 72 |
| CR1121 | 80 | 30 | 20 | 32 |
| CR1123 | 35 | 15 | 20 | 23 |
| CR550 | 0 | 100 | 0 | 19 |
| CR611 | 45 | 50 | 15 | 42 |
| CR644 | 60 | 17 | 40 | 89 |
| CR703 | 40 | 72 | 8 | 32 |
| CR719 | 30 | 62 | 8 | 51 |
| CR726 | 20 | 50 | 30 | 17 |

[a] % of tumor cells within tumor area

[b] % of necrosis/fibrosis within tumor area

[c] In silico tumor purity results using PureCN tool

patients from PureCN. As shown in Fig 6, we found no statistically significant difference between naïve and treated patients.

## Discussion

Determining tumor purity is important because of the role that stroma plays in cancer progression and because of the confounding effect low purity has on molecular analyses of the

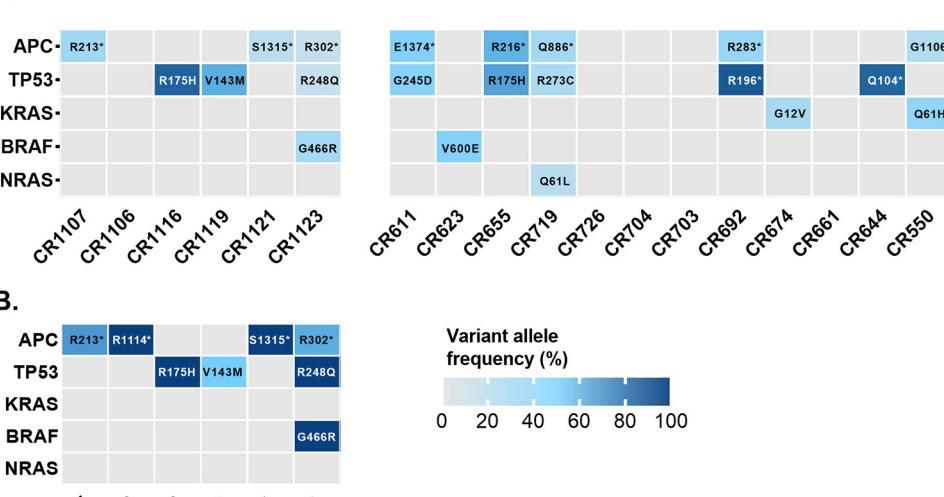

**Fig 4.** Variant allele frequencies of the pathogenic single nucleotide variants detected in CRCLM tumors (**A**) from 18 patients and in tumor organoids (**B**) derived from 6 of these patients. PDO; patient-derived tumor organoids, PDXO; patient-derived xenograft organoids. Amino acid change of each variant is displayed within the heatmap cells.

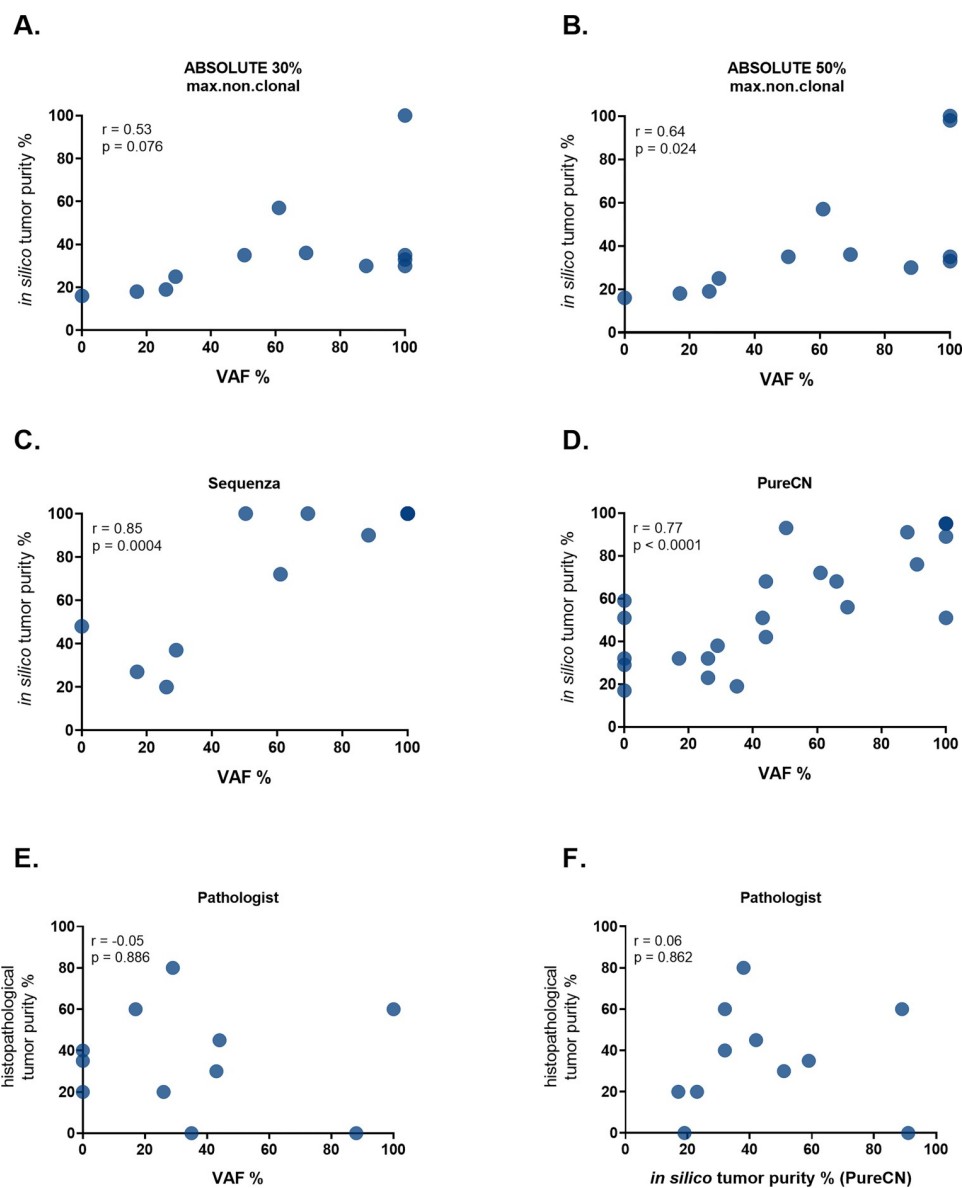

**Fig 5. Correlation between the variant allele frequency (VAF) of the most prevalent pathogenic variant in each sample and of *in silico* and pathologist tumor purity results. (A)** and (**B**) **ABSOLUTE, (C) Sequenza, (D) PureCN**, or (**E.**) pathologist tumor purity assessment. (**F**) Correlation between the PureCN *in silico* tumor purity results and pathologist purity assessment. Each dot represents a patient tumor specimen or a tumor-derived organoid sample.

malignant cells. Here we determined tumor purity in colorectal cancer liver metastasis specimens from 18 patients as well as in matching tumor organoids from 6 patients. By using whole-exome sequencing data and three *in silico* tumor purity tools, ABSOLUTE, Sequenza, and PureCN, we found lower tumor purity than has previously been reported for CRC [8]. This may be specific for liver metastases as previous reports have focused on primary CRC tumors. However, the lower median purity estimates we observed might also be due to the unbiased selection of the tumors we employed in this study as opposed to studies using the TCGA database which is composed mainly of samples from patients without prior chemotherapy treatment and high tumor purity as determined by histopathological evaluation [20].

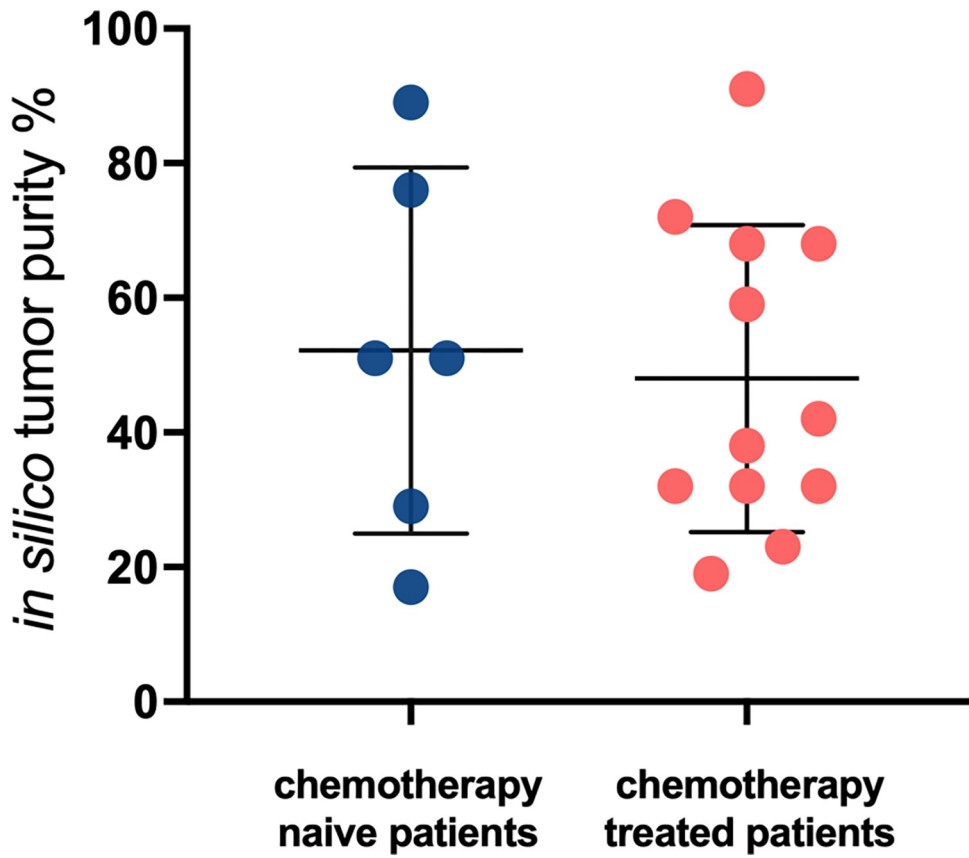

**Fig 6. PureCN tumor purity estimates with mean and standard deviation for all 18 patients grouped by their chemotherapy status prior to hepatectomy.**

Pathogenic mutations in APC, KRAS and TP53, are typically found in 80%, 40–50% and 70% of metastatic CRC tumors, respectively [21, 22]. We found only a few of the common pathogenic variants in CRC and detected KRAS mutations in only 2 out of 18 (11.8%) samples. Additionally, VAFs of the mutations detected were low, meaning that only a minority of the sequencing reads harbored the variant while most were wildtype. In contrast, most of the detected mutations in patient tumors exhibited high VAFs in matching organoids. This was expected as protocols used to generate organoids generally enrich for rapidly growing epithelial cells and deplete stromal components [18, 19]. Overall, the VAF data agreed with the output of *in silico* purity tools, particularly of Sequenza and PureCN, both of which suggested low tumor purity for this CRCLM cohort.

The computational tools used in this study identified low median tumor purity across the sample set. Sequenza and PureCN gave similar results while ABSOLUTE produced lower estimates for most samples. Variation in results between Sequenza and PureCN may be attributed to the differences in copy number segmentation methods and the data utilized. PureCN relies on a database of normal samples and on the matched normal sample whereas Sequenza compares tumor data only to the matched normal sample [7]. Since the normal database used in this study consisted of normal samples collected from only 6 patients, PureCN's algorithm may have performed sub-optimally. In general, Sequenza and PureCN provide more informative output than ABSOLUTE. While ABSOLUTE only provided a selection of possible solutions, Sequenza and PureCN provided segmentation output, and several additional

visualizations to better inform the user of how their optimal solution was chosen. With ABSO-LUTE we observed that the max.non.clonal parameter greatly affected the tumor purity estimate. A default setting of 5% often did not produce an output for our samples which may indicate a high level of cancer cell heterogeneity. Attempting to take sub-clonality into account, we set the max.non.clonal parameter at 30% or 50% and obtained purity estimates for all samples. However, these were consistently lower than with the other computational tools. PureCN was the only tool that estimated purity from tumor only samples with no matching normal data available. We found significant positive correlation between the VAF results and the tumor purity estimates from all three computational tools. However, the correlation was strongest with Sequenza and PureCN. When analyzing mixed samples with known ratios of normal and tumor DNA, all tools underestimated purity. Notably, ABSOLUTE failed to predict high purity for a sample that consisted of 75% cancer DNA and resulted in a purity estimate of 26%, irrespective of the algorithm's max.non.clonal parameter.

For part of the samples, similar tumor purity estimates were observed when comparing results from *in silico* analysis and from the pathologist. However, we did not find statistically significant correlation between the results of the pathologist and the output of the computational tools. This lack of correlation has been reported by others [6, 7] and was suggested to result from the qualitative nature of pathologist estimates and the failure of the assessed slides to fully account for tumor heterogeneity. For our sample set this lack of correlation might also be explained by the sample evaluated by the pathologist not being from the same area of the tumor used to extract genomic DNA for WES. The only exception to this was the tumor sample from patient CR1107 for which the same FFPE sample was used for both DNA extraction and sectioning. Despite this, the *in silico* and pathologist purity estimates for this tumor were disconcordant. Establishing a standard whereby pathological and molecular samples are derived from adjacent tumor pieces may produce better agreement between these two types of purity analyses. In addition, a combined laser-capture microdissection and genomic measure might be employed to further yield accurate estimates of tumor purity. Importantly, we cannot discount the fact that pathologist was also able to provide estimates of necrosis and fibrosis in addition to tumor purity, parameters not currently available with the *in silico* tools.

Neoadjuvant chemotherapy has been reported to result in enrichment of cancer-associated fibroblasts within the residual tumor mass [23]. We noted a slightly lower mean purity i.e., higher stromal content, in tumors from patients who had received neoadjuvant chemotherapy prior to hepatectomy. However, this finding was not statistically significant. Additional specimens, with information on the type and the timing of the neoadjuvant therapy prior to surgery, are needed to investigate the effect of chemotherapy on tumor purity.

Our data shows that metastatic CRC tumors often have an unappreciated abundance of stromal cells that genomic and transcriptomic studies of prescreened databases such as TCGA underestimate. Further research is needed to evaluate whether this is a feature of liver metastases or if found across all stages of CRC tumors. We found considerable variation between tumor purity estimates from different in silico tools as well as from pathologist estimates. Therefore, molecular assays, both genomic and transcriptomic, and in silico tools should be employed together with histopathological assessment to estimate tumor purity more accurately.

## Author Contributions

**Conceptualization:** Eveliina Karelehto.

**Data curation:** Noura Tbeileh, Eveliina Karelehto.

**Formal analysis:** Noura Tbeileh, Aras N. Mattis, Eveliina Karelehto.

**Funding acquisition:** Robert S. Warren.

**Investigation:** Noura Tbeileh, Aras N. Mattis, Eveliina Karelehto.

**Methodology:** Luika Timmerman, Kan Toriguchi, David B. Donner, Eveliina Karelehto.

**Project administration:** Eveliina Karelehto.

**Resources:** Yosuke Kasai, Carlos Corvera, Eric Nakakura, Kenzo Hirose, Robert S. Warren.

**Software:** Noura Tbeileh.

**Supervision:** David B. Donner, Robert S. Warren, Eveliina Karelehto.

**Visualization:** Noura Tbeileh, Eveliina Karelehto.

**Writing – original draft:** Noura Tbeileh, Eveliina Karelehto.

**Writing – review & editing:** Luika Timmerman, Aras N. Mattis, Kan Toriguchi, Yosuke Kasai, Carlos Corvera, Eric Nakakura, Kenzo Hirose, David B. Donner, Robert S. Warren.

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
