## [Decision Letter · Decision Letter 0]

17 Feb 2022

PONE-D-21-40240Metastatic colorectal adenocarcinoma tumor purity assessment from whole exome sequencing dataPLOS ONE

Dear Dr. Timmerman,

Thank you for submitting your manuscript to PLOS ONE. After careful consideration, we feel that it has merit but does not fully meet PLOS ONE’s publication criteria as it currently stands. Therefore, we invite you to submit a revised version of the manuscript that addresses the points raised during the review process.

Please address all of the reviewers comments, and consider expanding upon the Data Availability Statement to clarify what criteria researchers must meet to obtain the WES data. Alternatively, consider depositing the WES data in a repository for controlled access data.

We look forward to receiving your revised manuscript.

Kind regards,

Elizabeth Christie

Academic Editor

PLOS ONE

Journal Requirements:

Reviewers' comments:

Reviewer's Responses to Questions

**Comments to the Author**

1. Is the manuscript technically sound, and do the data support the conclusions?

Reviewer #1: Yes

Reviewer #2: No

2. Has the statistical analysis been performed appropriately and rigorously? 

Reviewer #1: Yes

Reviewer #2: Yes

3. Have the authors made all data underlying the findings in their manuscript fully available?

Reviewer #1: Yes

Reviewer #2: No

4. Is the manuscript presented in an intelligible fashion and written in standard English?

Reviewer #1: Yes

Reviewer #2: Yes

5. Review Comments to the Author

Reviewer #1: Tbeileh et al. studied the proportion of stromal cells in hepatic CRC metastases by performing computational tumor purity analysis based on whole exome sequencing data. They used an unbiased inhouse collection of tumor specimens. Matching tumor-derived organoids were also utilized as a mostly pure cancer control. They applied three in silico tumor purity tools, ABSOLUTE, Sequenza and PureCN. They found that patient metastasis specimens had a median tumor purity below 50%. In line with this, some of the VAFs of oncogenes and tumor suppressor genes were undetectable or low in the patient tumors, although positive correlation was observed between VAFs and in silico tumor purity estimates. Sequenza and PureCN produced concordant results whereas ABSOLUTE yielded lower purity estimates for all samples.

Focusing on the metastatic CRC with non-biased samples, and showing the lower cancer cell purity than expected are interesting findings. There are concerns as follow.

Comments

The platforms of tumor purity estimation in this study are relatively old. Please explain why they applied these three methods among many others.

Doi: 10.1093/bioinformatics/btz406.

DOI: 10.1200/PO.20.00016 JCO

Organoids were used as a control of mostly pure cancer cells in this study. Since it is difficult to understand the context, it should be briefly noted in abstract why the organoids were used.

Method for organoid preparation and culture should be described more in detail, especially the timing of the sampling after preparation or culture. Longer culture period might select the cancer cells adapted to the culture conditions.

The timings of the operations after chemotherapy should be shown.

Histopathological analysis should be described more in detail, since it will make a big difference when the estimation is based on the tumor/stromal ‘area’ or ‘cells’. It would be better to show representative HE images along with the instruction of the way of the histopathological analysis; each area, each cell, and so on. It would be better to perform the histopathological analysis at multiple sites to show the heterogenous cancer cell purity in one tumor.

P3L42 Since the molecular/computational and histopathological estimation are not consistent in this study, the conclusion of the abstract, just saying combination of the analyses is important is not appropriate.

P6L91-93

Please describe the relationship between snap frozen samples and fresh samples. Were they from the same samples or from different sites?

P10L180

‘tumor’ is better to be ‘cancer cells’

P11-P12

The purity estimation results are described as ‘low’ and ‘high’. Please make the criteria of low and high clearer.

Table 2

Since this table is not in comparison with the other types of estimation, it is difficult to interpret the data for a reader. It would be better to show the tumor purity at least also in figure 1.

It seems like that the histopathological estimation is performed by areas but not cells. If so, it is not surprising that the estimation by ‘area’ are not consistent with the estimation by ‘cells’.

In addition to CR1107, CR1121 is also consistent with in silico estimation.

Figure 1C

ABSOLUTE might yield lower purity estimates for higher purity samples (figure 1B), although when the purity is lower, it might be superior than others (figure 1C). Therefore, the conclusion ‘ABSOLUTE yielded lower purity estimates for all samples’ might not be appropriate.

P14L249, Figure 5

It is hard to say ‘a trend’ from this data.

Figure 3

Please explain the discrepancy of APC mutation of CR1106 between A and B

Reviewer #2: In this manuscript Tbeilah et al, explore the use of three in silico tools which utilise WES data to assess the tumour purity in liver mets associated with colorectal cancer. Whilst the premise is a good one, the limited number of samples overall, and the limited number with pathology review make it difficult to draw significant conclusions. I recommend that the authors revise the manuscript and incorporate additional samples in their analysis.

1.Pathologist assessment was performed on only 6 cases. This sample size is rather small and given the inconsistencies in the performance of the 3 tools it is very difficult to make any conclusions. This aspect of the work would be strengthened with a larger number of samples. This would then allow the authors to confirm (or not) and the trends described.

2.The in silico tumour purity vales for Sequenza and PureCN was similar for most samples but was very different for cases PDXO1107 and PDXO1123. It is reasonable to assume (as also mentioned by the authors) that organoid samples which have been enriched for epithelial cells would have a reasonably high purity. Can the authors comment on this difference and explain the low purity levels estimated by PureCN and comment on the accuracy of this tool?

3. Sample CR1106 was estimated to have 3% tumour and 62% stroma by the pathologist. This was the lowest of all the samples. The VAF for APC of the associated organoid was very high but was not detected at all in the original tumour. Do the authors believe that there is a minimum purity threshold that needs to be met for the tools to be useful?

4. Fig 4 shows the correlation b/w the VAF of the most prevalent pathogenic variant for each sample and the in silico tumour purity results. Can the authors provide details of the what the variant was for each sample?

5. Can the authors explain why they chose to use a mix of PDO’s (n=2) and PDXO’s (n=4)? Again, these numbers are small and introduce additional variables.

6. PLOS authors have the option to publish the peer review history of their article (what does this mean?). If published, this will include your full peer review and any attached files.

Reviewer #1: No

Reviewer #2: No

---

## [Author Response · Author response to Decision Letter 0]

12 May 2022

We have uploaded a reviewer response document, a revised manuscript with and without the changes tracked, and a new set of figures of proper format per the editor's request. We have no changes to make to the financial disclosures. The data availability statement has been changed to indicate that the data will be housed at dbGaP for limited general research use (GRU). Some limitation is necessary since our data is whole exome sequences, which, with some effort, may be used to identify an individual.

Our manuscript meets PLOS ONE's style requirements, and my ORCID number is correct. We added a section at the end of the methods section of the manuscript titled "Data Availability" which states that the data is (will be) housed at the NIH web site named dbGaP, per an email from Radovan Lumban-Tobing, and another from Edrian Nim Tolentino. Per their request we have also added consent information into the manuscript.

Specific reviewer's comments are addressed in the document "Response to Reviewers", uploaded today.

---

## [Decision Letter · Decision Letter 1]

31 May 2022

PONE-D-21-40240R1Metastatic colorectal adenocarcinoma tumor purity assessment from whole exome sequencing dataPLOS ONE

Dear Dr. Timmerman,

Thank you for submitting your manuscript to PLOS ONE. After careful consideration, we feel that it has merit but there are some minor points raised by Reviewer #1 to be addressed. Therefore, we invite you to submit a revised version of the manuscript that addresses the points raised during the review process.

We look forward to receiving your revised manuscript.

Kind regards,

Elizabeth Christie

Academic Editor

PLOS ONE

Journal Requirements:

Reviewers' comments:

Reviewer's Responses to Questions

**Comments to the Author**

1. If the authors have adequately addressed your comments raised in a previous round of review and you feel that this manuscript is now acceptable for publication, you may indicate that here to bypass the “Comments to the Author” section, enter your conflict of interest statement in the “Confidential to Editor” section, and submit your "Accept" recommendation.

Reviewer #1: (No Response)

Reviewer #2: All comments have been addressed

2. Is the manuscript technically sound, and do the data support the conclusions?

Reviewer #1: Yes

Reviewer #2: (No Response)

3. Has the statistical analysis been performed appropriately and rigorously? 

Reviewer #1: Yes

Reviewer #2: (No Response)

4. Have the authors made all data underlying the findings in their manuscript fully available?

Reviewer #1: Yes

Reviewer #2: (No Response)

5. Is the manuscript presented in an intelligible fashion and written in standard English?

Reviewer #1: Yes

Reviewer #2: (No Response)

6. Review Comments to the Author

Reviewer #1: I should have mentioned them in the first comments, but I hope the following comments will help to make the manuscript better.

Abstract

1) Overall, metastatic specimens had a median tumor purity below 50% whereas the organoids were enriched for cancer cells with purity estimates above 90%.

As the author added in the revised manuscript, adding ‘according to all computational methods’ would be better than just ‘overall’. Where is the value of ‘90%’ described in the results? Actually, the average of Figure 1B is 70%.

2) In line with this, VAFs of oncogenes and tumor suppressor genes were undetectable or low in patient tumors, but high in matching organoid cultures.

Since some of the VAFs are 100% in Figure 4, it would be better to put ‘in some cases’ in this sentence.

3)Figure

In Figure 5F, the legend of the x-axis, in silico tumor purity % (PureCN) would be better than just PureCN.

Reviewer #2: (No Response)

7. PLOS authors have the option to publish the peer review history of their article (what does this mean?). If published, this will include your full peer review and any attached files.

Reviewer #1: No

Reviewer #2: No

---

## [Author Response · Author response to Decision Letter 1]

15 Jun 2022

Response to Reviewers' comments for the manuscript: Metastatic colorectal adenocarcinoma purity assessment from whole exome sequencing

Reviewer #1: I should have mentioned them in the first comments, but I hope the following comments will help to make the manuscript better.

1. Overall, metastatic specimens had a median tumor purity below 50% whereas the organoids were enriched for cancer cells with purity estimates above 90%. 

As the author added in the revised manuscript, adding ‘according to all computational methods’ would be better than just ‘overall’. Where is the value of ‘90%’ described in the results? Actually, the average of Figure 1B is 70%.

We thank the reviewer for pointing out this error. We neglected to update this part of the abstract when we re-analyzed data for our previous resubmission. We have now modified the abstract text to show the purity median for all patient samples and for all organoids (across all computational methods), line 37. Median tumor purity results are given separately for each computational tool in the results section. Thus we have harmonized the description of the analysis results in the abstract with the analysis in the body of the manuscript.

2. In line with this, VAFs of oncogenes and tumor suppressor genes were undetectable or low in patient tumors, but high in matching organoid cultures.

Since some of the VAFs are 100% in Figure 4, it would be better to put ‘in some cases’ in this sentence.

We are unsure where in this sentence the reviewer wishes to include ‘in some cases’. We slightly modified the sentence in the abstract and hope it is clearer now, line 40.

3. In Figure 5F: the legend of the x-axis, in silico tumor purity % (PureCN) would be better than just PureCN.

We have modified the figure legend as suggested.

---

## [Decision Letter · Decision Letter 2]

29 Jun 2022

Metastatic colorectal adenocarcinoma tumor purity assessment from whole exome sequencing data

PONE-D-21-40240R2

Dear Dr. Timmerman,

We’re pleased to inform you that your manuscript has been judged scientifically suitable for publication and will be formally accepted for publication once it meets all outstanding technical requirements.

Kind regards,

Elizabeth Christie

Academic Editor

PLOS ONE

Additional Editor Comments (optional):

Reviewers' comments:

Reviewer's Responses to Questions

**Comments to the Author**

1. If the authors have adequately addressed your comments raised in a previous round of review and you feel that this manuscript is now acceptable for publication, you may indicate that here to bypass the “Comments to the Author” section, enter your conflict of interest statement in the “Confidential to Editor” section, and submit your "Accept" recommendation.

Reviewer #1: All comments have been addressed

2. Is the manuscript technically sound, and do the data support the conclusions?

Reviewer #1: Yes

3. Has the statistical analysis been performed appropriately and rigorously? 

Reviewer #1: Yes

4. Have the authors made all data underlying the findings in their manuscript fully available?

Reviewer #1: Yes

5. Is the manuscript presented in an intelligible fashion and written in standard English?

Reviewer #1: Yes

6. Review Comments to the Author

Reviewer #1: All comments have been well addressed.

7. PLOS authors have the option to publish the peer review history of their article (what does this mean?). If published, this will include your full peer review and any attached files.

Reviewer #1: No

---

## [Editor Report · Acceptance letter]

28 Mar 2023

PONE-D-21-40240R2 

Metastatic colorectal adenocarcinoma tumor purity assessment from whole exome sequencing data 

Dear Dr. Timmerman:

I'm pleased to inform you that your manuscript has been deemed suitable for publication in PLOS ONE. Congratulations! Your manuscript is now with our production department. 

Kind regards, 

on behalf of

Dr. Elizabeth Christie 

Academic Editor

PLOS ONE